# Horizontal gene transfer and ecological interactions jointly control microbiome stability

Katharine Z. Coyte[1]*, Cagla Stevenson[2], Christopher G. Knight[3], Ellie Harrison[2], James P. J. Hall[4], Michael A. Brockhurst[1]*

1 Division of Evolution and Genomic Sciences, Faculty of Biology, Medicine and Health, University of Manchester, Manchester, United Kingdom, 2 Department of Animal and Plant Sciences, The University of Sheffield, Sheffield, United Kingdom, 3 Department of Earth and Environmental Sciences, Faculty of Science and Engineering, University of Manchester, Manchester, United Kingdom, 4 Department of Evolution, Ecology and Behaviour, Institute of Infection, Veterinary and Ecological Sciences, University of Liverpool, Liverpool, United Kingdom

* katharine.coyte@manchester.ac.uk (KZC); michael.brockhurst@manchester.ac.uk (MAB)

**Data Availability Statement:** All code and raw data underlying this work can be found at https://github.com/katcoyte/hgt-microbiome-stability/.

## Abstract

Genes encoding resistance to stressors, such as antibiotics or environmental pollutants, are widespread across microbiomes, often encoded on mobile genetic elements. Yet, despite their prevalence, the impact of resistance genes and their mobility upon the dynamics of microbial communities remains largely unknown. Here we develop eco-evolutionary theory to explore how resistance genes alter the stability of diverse microbiomes in response to stressors. We show that adding resistance genes to a microbiome typically increases its overall stability, particularly for genes on mobile genetic elements with high transfer rates that efficiently spread resistance throughout the community. However, the impact of resistance genes upon the stability of individual taxa varies dramatically depending upon the identity of individual taxa, the mobility of the resistance gene, and the network of ecological interactions within the community. Nonmobile resistance genes can benefit susceptible taxa in cooperative communities yet damage those in competitive communities. Moreover, while the transfer of mobile resistance genes generally increases the stability of previously susceptible recipient taxa to perturbation, it can decrease the stability of the originally resistant donor taxon. We confirmed key theoretical predictions experimentally using competitive soil microcosm communities. Here the stability of a susceptible microbial community to perturbation was increased by adding mobile resistance genes encoded on conjugative plasmids but was decreased when these same genes were encoded on the chromosome. Together, these findings highlight the importance of the interplay between ecological interactions and horizontal gene transfer in driving the eco-evolutionary dynamics of diverse microbiomes.

## Background

Diverse microbial communities colonize virtually every habitat on earth, shaping their abiotic environments and the health of their multicellular hosts [1–3]. Stably maintaining a diverse

**Funding:** C.S. was supported by an ACCE DTP NERC PhD studentship. M.A.B., J.P.J.H. and E.H. were funded by grants BB/R014884/1 and NE/R008825/1. The funders had no role in study design, data collection and analysis, decision to publish, or preparation of the manuscript.

**Competing interests:** The authors have declared that no competing interests exist.

**Abbreviations:** CI, credibility interval; gLV, generalized Lotka–Volterra; HGT, horizontal gene transfer; OTU, operational taxonomic unit.

microbial community is critical for overall microbiome performance, ensuring that the presence of beneficial species or desirable metabolic traits are retained over time [4–7]. In particular, it is crucial that microbial communities can robustly withstand perturbations caused by external stressors, such as environmental pollutants or antibiotics, which may otherwise dramatically reduce overall microbiome abundances and diversity [4,8–10]. Antibiotic-induced changes in community composition have been correlated with a range of adverse health outcomes in host-associated microbiomes [11], while losses in microbial diversity triggered by heavy metal and other toxic pollution have been linked to reduced nutrient cycling within environmental microbiomes [12]. Yet, despite the importance of withstanding perturbations, the forces shaping the stability of microbial communities remain poorly understood.

Existing theoretical work on microbiome stability has focused primarily on the role of ecological factors, developing mathematical models to disentangle how forces such as microbe–microbe interactions or different classes of stressors influence how microbiomes respond to perturbations [13,14]. However, such models have typically assumed that all species within a given microbiome are equally affected by these stressors. Perhaps more importantly, these models typically also assume microbial species remain equally susceptible to stressors over time. In practice, antibiotic or toxin resistance genes are prevalent within microbial communities, often encoded on mobile genetic elements such as plasmids or temperate phages, which can rapidly spread within and between microbial species by horizontal gene transfer (HGT) [15–17]. Therefore, not only are species within microbiomes differentially impacted by stressors, but the rapid spread of mobile genetic elements may dynamically alter the susceptibility of individual microbes to these stressors over short periods of time. These resistance genes and their mobility are highly likely to influence overall microbiome stability, yet exactly how remains unknown.

Here we develop eco-evolutionary theory to examine how the presence and mobility of resistance genes within microbial communities shape microbiome stability in the face of stressors. We then test our key predictions using model soil microbiomes exposed to heavy metal perturbations. In general, our modelling predicts that resistance genes increase overall microbiome stability, with this beneficial effect increasing with increasing gene mobility. However, we also find that a resistance gene can have very different impacts on individual community members, depending upon the precise balance of ecological interactions within a given microbiome, and the mobility of the resistance gene itself. Immobile resistance genes may benefit susceptible species in cooperative communities yet damage those in competitive communities. Meanwhile, though the spread of mobile resistance genes tends to increase overall community stability, it can, counterintuitively, decrease the stability of the originally resistant species. Crucially, our experiments support these key predictions, confirming the beneficial impacts of resistance genes and their mobility on average community properties and recapitulating the adverse impacts of resistance genes on certain community members. Overall, our work highlights the critical importance of eco-evolutionary dynamics and HGT in shaping complex microbiomes.

## Results and discussion

### Mathematical model of eco-evolutionary microbiome dynamics

To understand the effect of resistance genes and their mobility on microbial community dynamics, we developed a simple and generalizable mathematical model of microbiome dynamics, built around the generalized Lotka–Volterra (gLV) equations (Fig 1A). As in previous work [14,18–21], our model assumes the growth of each taxon within a microbiome is determined by the combination of its own intrinsic growth rate ($r_i$), its competition with kin

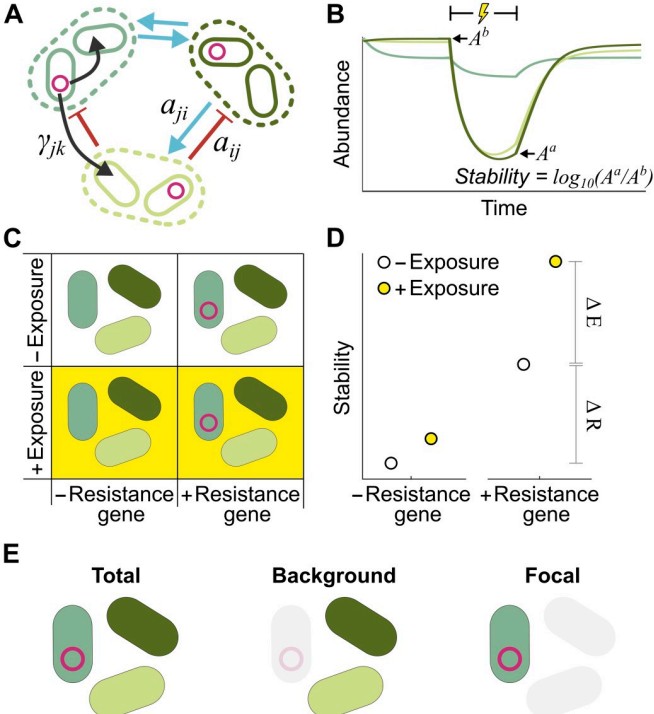

**Fig 1. Mathematical modelling captures eco-evolutionary dynamics of microbial communities. (A)** Schematic illustrating our mathematical model; each taxon (dashed line) is composed of 2 populations, with and without a resistance gene. Species impact one another's growth through ecological interactions such as cooperation or competition (blue and red arrows), while horizontal gene transfer enables resistance genes to spread within and between taxa (black arrows). **(B)** Schematic illustrating representative microbiome dynamics, capturing microbial dynamics before, during, and after an external perturbation (lightning bolt). We calculate each taxon's abundance immediately before, $A^b$, and after, $A^a$, the perturbation period, then define stability as the average logged fold change for each taxon (mean $\log_{10}(A^a/A^b)$). **(C)** Schematic illustrating our 4 modelling scenarios: communities with and without resistance genes, and with and without prior exposure to low-level stressors. **(D)** Comparing the 4 scenarios allows us to calculate the change in microbiome stability that results from the initial presence of a resistance gene ($\Delta R$), and the change in stability that results from prior exposure to low-level selection ($\Delta E$). **(E)** To disentangle the impact of resistance and selection on different taxa, we calculate $\Delta R$ and $\Delta E$ for the total community, the background community only, and the focal taxon alone.

($s_i$), and the combination of any interactions each taxon has with other community members ($a_{ij}$). Although simple, these gLV models have been shown to well capture and predict the dynamics of both host-associated and environmental microbial communities [14,18–21]. However, previous work on the stability of these communities in the face of perturbations has assumed all taxa within any given microbiome are equally impacted by environmental perturbations such as antibiotics [13,14]. That is, microbiome stability has typically been assessed by examining how communities respond to uniform, instantaneous changes in each constituent taxon's abundance. As our goal is to explore dynamic variability in the impact of stressors on individual microbial taxa owing to resistance genes, we now extended this basic model to explicitly incorporate a stressor that inhibits (or kills) susceptible cells and a potentially mobile resistance gene that protects cells encoding it, but at the cost of a reduced intrinsic growth rate. This adjusted model allowed us to assess microbiome stability in the face of perturbations when the susceptibility of individuals to those perturbations varies between taxa and dynamically over time.

Using this model, we could simulate the individual taxa abundances of any given microbiome over time. More specifically, we could simulate the scenario in which a given microbiome first has a fixed period of time to adjust to a new environment and is then briefly exposed to an external stressor such as a heavy metal or an antibiotic perturbation (Fig 1B). By measuring the change in each taxon's abundance during this perturbation, we could thereby quantify the stability of that microbiome. More specifically, we focused primarily on one key measure of stability: the average decrease in taxa abundances following a perturbation, often termed community Robustness (Fig 1B). This measure allowed us to quantify the immediate response of the community to a perturbation. However, for completeness, we also quantified 2 further metrics of stability—the change in community composition induced by the perturbation (measured as Bray–Curtis dissimilarity), and the time taken for the community to return to its original state following the perturbation—each of which produced qualitatively similar results (see S1 Text).

Having established this basic model, we used it to explore the impact of resistance genes on the stability of microbiome communities. Specifically, we generated a series of diverse multi-taxa microbial communities, then quantified the stability of each of these microbiomes under 4 distinct scenarios (Fig 1C). First, we simulated microbiome dynamics when all taxa were susceptible to the stressor, and then again when a randomly chosen focal taxon carried a gene encoding resistance to the stressor. Depending upon its mobility, this resistance gene could spread into susceptible cells during both the initial adjustment period and the perturbation window. Next, we repeated this process but allowed each microbiome to first adjust to the presence of a low level of the stressor, for example, simulating prior exposure to subinhibitory levels of antibiotics or pollutants. This process allows us to define 2 metrics: the change in microbiome stability resulting from the initial presence of a resistance gene, ΔR, and the change in microbiome stability resulting from prior exposure to low-level selection, ΔE (Fig 1D). A positive ΔR indicates that a resistance gene increases stability, and a negative ΔR indicates that the resistance gene decreases stability (and equivalently for ΔE in terms of the effect of prior exposure upon stability).

Crucially, for each community, we calculated ΔR and ΔE for the whole microbiome (Fig 1E), the focal taxon only (that is, the taxon that originally carried the resistance gene), and the background community only (that is, all taxa except the focal taxon). This enabled us to disentangle the impact of the resistance gene on microbiome as a whole from its impact on the initially susceptible and initially resistant compartments of the microbiome community separately. Using this modelling framework, we could then explore in depth how the presence of resistance genes in an individual species impacts microbiome dynamics and stability overall and in defined compartments of the microbiome.

## Mobile resistance genes increase stability of noninteracting communities

We began by exploring how the presence and mobility of resistance genes influenced the stability of microbiomes in the absence of intertaxa ecological interactions (that is, all $a_{ij} = 0$). To do so, we generated a set of communities with and without resistance genes. We then systematically varied the ability of these resistance genes to transfer within and between taxa (Fig 2A), capturing all degrees of gene mobility from immobile (e.g., a chromosomally encoded resistance gene) to highly mobile (e.g., a resistance gene encoded by a highly conjugative and promiscuous plasmid).

Any resistance gene increased overall microbiome stability, decreasing the average drop in community abundances following the onset of the perturbation (Fig 2B, ΔR > 0). However, examining background and focal taxa separately revealed that, in most cases, this increase was

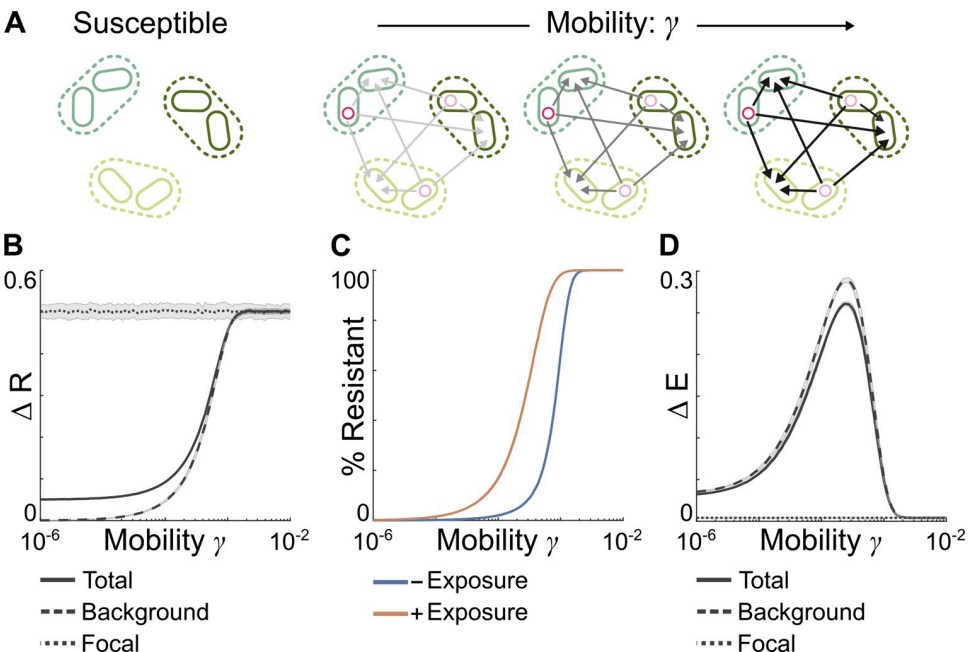

**Fig 2. In simple communities, mobile resistance genes increase microbiome stability. (A)** Schematic illustrating our modelling approach. We generate a set of simple microbiomes without interactions between taxa, then for each community, we simulate the effect of a series of resistance genes with increasing mobility. **(B)** ΔR, the change in stability resulting from the initial presence of a resistance gene, for the whole community (solid line), the background community (dashed line), and the focal taxon (dotted line). Any resistance gene increases overall community stability, but only highly mobile resistance genes substantially increase the stability of background taxa. **(C)** Resistance gene frequency immediately before the perturbation within the background community with (orange) and without (blue) prior exposure to low-level stressor. Prior exposure increases resistance gene frequency, with this increase greatest for resistance genes with intermediate mobility. Only at extreme levels of resistance gene mobility does resistance fully saturate the background community. **(D)** ΔE, the change in stability resulting from prior exposure to low stressor levels, for the community as a whole (solid line), the background community (dashed line), and the focal taxon (dotted lined). Prior low-level selection increases the stability of both the community as a whole and background taxa, with this effect greatest for communities with intermediate mobility resistance genes. Throughout lines and shaded errors represent mean and standard deviation over 100 independent, 10-species communities, with model parameters given in Table 1. Underlying data at https://github.com/katcoyte/hgt-microbiome-stability.

driven solely by the increased stability of the focal taxon. That is, the presence of a resistance gene in the focal species drove up the average stability of the community as a whole, but in most cases, the stability of background taxa remained effectively unchanged (Fig 2B). To substantially increase the stability of the background microbiome, we found that the novel resistance gene must be highly mobile. This was because, prior to the perturbation, the resistance gene did not confer any benefit and thus could only spread into the background community when its transfer rate exceeded its rate of decline caused by negative selection against its cost (Fig 2C). However, low-mobility resistance genes could increase background taxa stability provided additional forces enhanced the spread of resistance prior to any perturbation. For example, prior exposure to low-level stressor selection introduced a weak benefit to harboring the resistance gene prior to the perturbation, enabling the mobile genetic element encoding the resistance gene to spread and reach low but nonzero frequencies in background community even at lower rates of gene mobility (Fig 2C). As a consequence, prior exposure substantially increased the stability of background species (Fig 2D, ΔE > 0). Notably, this effect was strongest for intermediate mobility resistance genes, as highly mobile resistance genes spread within the population even without prior exposure, while low-mobility genes remain relatively limited within the background population even with prior exposure.

## Intertaxa interactions modulate the impact of resistance genes on microbiome stability

Having established these baseline properties of the system, we next examined how the impact of resistance genes upon stability is modulated by interactions between taxa. Specifically, we allowed individual taxa within our simulated communities to interact with one another in a variety of different ways, ranging from competition and ammensalism (−/− and −/0 interactions, respectively), through exploitation (+/−), to cooperation and commensalism (+/+ and +/0). We then generated a range of different microbial communities, systematically varying the proportion of each interaction type (Fig 3A). As previously, we then simulated the effect of resistance genes on these communities, also systematically changing the mobility of the resistance gene.

As in noninteracting communities, any resistance gene typically increased average overall community stability, and this increase was higher for more mobile resistance genes (Fig 3B and Figs A-E in S1 Text). However, this beneficial effect of resistance genes varied with interaction type and was far stronger in microbiomes with a high proportion of cooperative interactions. In these cooperative communities, individual taxa benefited both directly from acquiring resistance genes, and indirectly from their cooperative partners acquiring resistance,

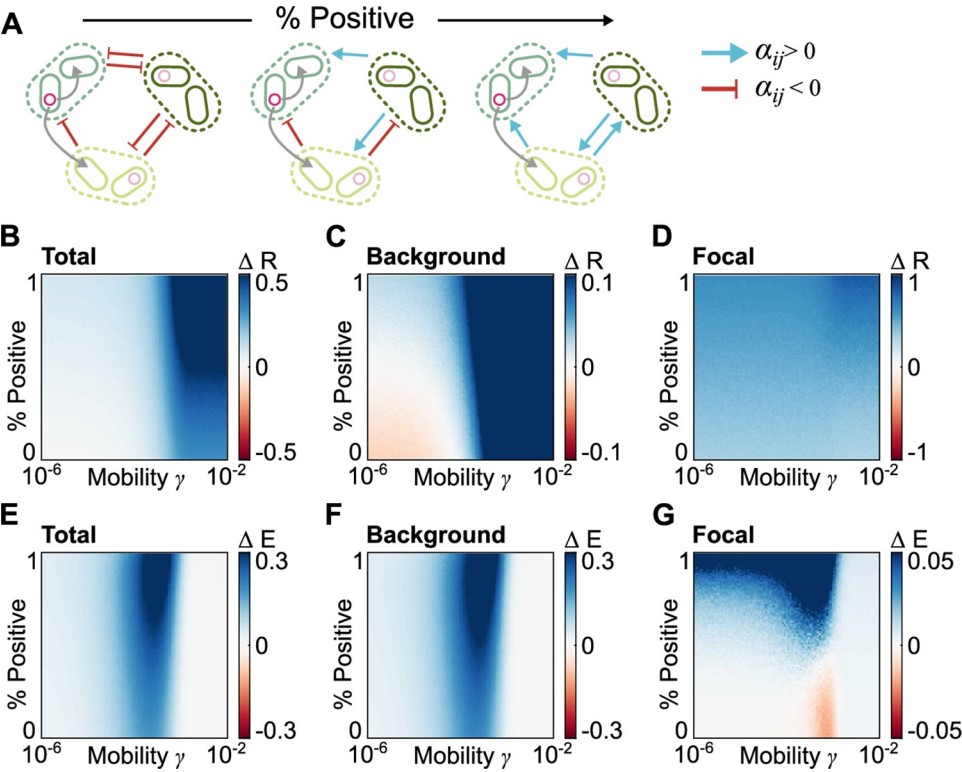

**Fig 3. Interactions between taxa modulate the effect of resistance genes. (A)** Schematic illustrating our modelling approach. We generated a series of diverse microbiomes, systematically increasing the frequency with which microbes facilitate one another's growth. **(B-D)** Average ΔR, the change in stability resulting from the initial presence of a resistance gene, under varying community types and resistance gene transfer rates, shown for the whole microbiome (**B**), the background community (**C**), and the focal taxon alone (**D**). **(E-G)** Average ΔE, the change in stability resulting from prior exposure to low stressor levels, under varying community types and resistance gene transfer rates, shown for the whole microbiome (**E**), the background community (**F**), and the focal taxon alone (**G**). Throughout, patch color represents mean ΔR or ΔE over 100 independent, 10-taxa communities, across a range of 101 Positivity and $\gamma$ values. Other model parameters given in Table 1, underlying data at https://github.com/katcoyte/hgt-microbiome-stability.

which, in turn, helped to buffer the negative impact of the stressor perturbation. The principal effect of prior low-level stressor exposure was once again to reduce the level of gene mobility required for resistance genes to spread within the community. And, as a consequence, this prior stressor exposure increased the stabilizing effect of mobile resistance genes on overall community stability, across all interaction types (Fig 3E and Figs A-E in S1 Text, although, again, this effect was strongest for resistance genes with intermediate mobility).

Remarkably, however, while mobile resistance genes increased overall microbiome stability, examining focal and background taxa separately revealed radically different impacts of the resistance gene on individual taxa. That is, background and focal taxa showed markedly different responses to resistance genes depending upon the precise manner in which taxa were interacting and the mobility of the resistance gene (Fig 3C and 3D and Figs A-E in S1 Text). In cooperative microbiomes, background taxa benefited from the initial presence of a resistance gene regardless of its mobility. This occurred because, by promoting the survival of taxa with whom a susceptible taxon cooperates, resistance genes aid recovery of the susceptible taxon regardless of whether they have access to the resistance gene through HGT. However, in competitive communities, highly mobile resistance genes increased background community stability while low-mobility genes *reduced* background community stability (Fig 3C). That is, in highly competitive communities, most taxa were less stable when another member of the community harbored an immobile or low-mobility resistance gene than when all taxa were susceptible.

What drove this negative impact of resistance genes in competitive communities? In fully susceptible competitive communities, during a perturbation, every taxon experienced a reduction in their net growth rate, typically resulting in a decrease in their overall abundance. However, as a consequence, each taxon also benefited from some competitive release—that is, the negative impact of competitors was reduced as these competitors also decreased in abundance. In contrast, if one taxon acquired an immobile or low-mobility resistance gene, then this focal taxon remained at a high density during the perturbation—and as such, susceptible taxa suffered not only from the stressor-mediated inhibition, but also from continued strong competitive inhibition by the focal taxon. This stark difference in dynamics between the community as a whole and background taxa reveals the critical importance of looking beyond average community properties when studying microbiome stability. Meanwhile, the dramatic differences between community types underlines the vital role of ecological interactions in shaping microbiome dynamics.

This impact of competitive release also modulated the impact of prior low-level exposure—again with very different impacts on background and focal taxa. Background taxa benefited from prior exposure to low-level stressors regardless of community context (Fig 3F), because this prior exposure promoted the spread of mobile resistance into the background community. Moreover, this spread of mobile resistance also stabilized cooperative focal taxa, as these taxa now benefited from their cooperative partners acquiring resistance genes and thus remaining at high abundances during perturbations (Fig 3G). In certain competitive communities, however, prior selection could slightly reduce the stability of the focal taxon ($\Delta E < 0$; Fig 3G) because the spread of mobile resistance into background taxa meant that the focal taxon no longer benefited from any competitive release during perturbations. This effect was restricted to communities with intermediate mobility resistance genes, which were driven to high frequency in the background community by prior exposure but otherwise would not have spread to high frequency. Altogether, our results suggest mobile resistance genes can have a wide variety of effects, with the precise consequences depending upon which taxa are being examined, how they interact with one another, and the mobility of the resistance gene.

## Experimental microbial communities reproduce key theoretical predictions

To test our predictions about the impact of resistance genes and their mobility on stability, we developed an experimental model microbiome system. Specifically, we generated model microbiomes by inoculating sterile potting soil microcosms with 96 soil bacterial isolates (representing 14 unique operational taxonomic units (OTUs), our background community) and one focal taxon *Pseudomonas fluorescens* SBW25. In each microcosm, this focal species was either susceptible to mercury (Hg$^S$), or carried a mercury resistance operon, encoded either on the chromosome or on one of 2 conjugative plasmids, pQBR103 and pQBR57. Whereas the chromosomally encoded resistance is nonmobile, the conjugative plasmids can transfer mercury resistance to other taxa [22–24]. Having assembled these communities, we allowed them to adjust to their conditions for 10 serial transfers (40 days) either with or without low-level mercury exposure (Fig 4A). Communities were then perturbed with a pulse of high concentration mercury, then propagated without mercury for 2 additional serial transfers, mirroring the mode of perturbation used in our modelling framework. We determined the composition of bacterial communities by amplicon sequencing of the 16S rRNA gene before and after the

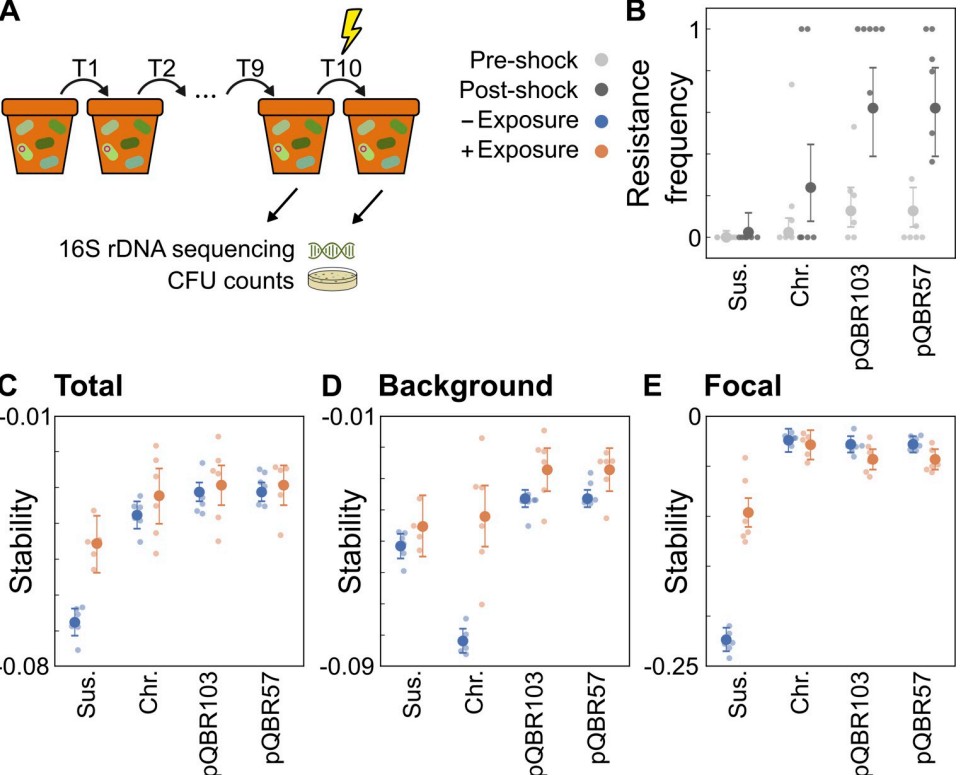

**Fig 4. Experimental microbiomes recapitulate the predicted impact of resistance genes on microbiome stability.** (**A**) Schematic illustrating our experimental system. Here soil microcosms are seeded with microbial communities, allowed to acclimatize with or without low-level mercury, then perturbed with a high-level mercury pulse. (**B**) Frequency of resistance within the background population before (light grey) and after (dark grey) the high-level mercury pulse in the absence of prior mercury exposure. Calculated for each experimental condition (fully susceptible, Sus., and Chromosomal, Chr. or plasmid-carried resistance, pQBR103, pQBR57). (**C**) Total microbiome stability, as measured by log10(Fold Change) in taxa abundances following perturbation, such that more negative values indicate a less stable community. (**D**) Stability of the background community individually. (**E**) Stability of the focal taxon alone. Throughout, orange and blue dots indicate community stability with and without prior mercury exposure, respectively, with each condition (resistance × prior selection) containing $n = 6$ independent samples. Underlying data at https://github.com/katcoyte/hgt-microbiome-stability.

perturbation and also estimated the abundances of the total community by colony counts. As in our models, we then quantified community stability as the change in each taxon's abundance between the pre- and post-perturbation samples.

After adjusting, each microbiome contained between 3 and 5 detectable taxa, namely, OTU_107 (the focal taxon, *P. fluorescens* SBW25), OTU_3 (*Pseudomonas* sp.), OTU_14 (*Pseudomonas umsongensis*), OTU_9 (*Bacillus megaterium*), and OTU_19 (*Bacillus simplex*) out of a total 14 OTUs present in the starting community. Although the exact ecological interactions occurring between each of these taxa are not known, there is good evidence for exploitative and interference competition occurring between *Pseudomonas* species [25,26] and between members of the *Bacillus* and *Pseudomonas* genera [27–30], suggesting this is likely to have been a competitive community. As such, our model predicts that an introduced resistance gene would be likely to increase overall microbiome stability but that background taxa would only benefit from plasmid-encoded resistance and might suffer from the introduction of chromosomally encoded genes. Moreover, our modelling also predicts that prior selection might slightly decrease the stability of the focal taxon when it harbored plasmid-encoded mercury resistance due to transfer of mobile resistance limiting the benefits of competitive release.

As predicted, the presence of any resistance gene in the focal taxon SBW25 significantly increased overall microbiome stability, strongly reducing the decline in abundance caused by the mercury pulse (Fig 4C; effect of resistance = 0.030 [0.025 to 0.035, 95% CI], Bayesian linear model BLM1; see S1 Text). Similarly, as predicted, there was an overall positive effect on stability of prior exposure to low-level mercury selection (effect of exposure = 0.022 [0.013 to 0.031, 95% CI], BLM1). We also observed a positive impact of gene mobility on overall microbiome stability; however, this effect was small (estimated effect of mobility on stability, given resistance = 0.0065 [0.0017 to 0.011, 95% CI], BLM1) and notably did not change significantly with prior exposure (estimated interaction effect between exposure and mobility, given resistance = −0.0035 [−0.014 to 0.0075, 95% CI], BLM1), suggesting that the large increase in total community stability conferred by the introduction of a resistance gene into the focal taxon masks the subtler changes in stability conferred by the transfer of resistance genes into the background community. Together, these results confirmed theoretical predictions that resistance genes increase stability at the scale of entire communities, but such measures are likely to ignore important differences between individual taxa and compartments of the microbiome.

In contrast, and as predicted by our mathematical modelling, calculating the stability of background and focal taxa separately revealed markedly different behavior between taxa, and a far stronger effect of resistance gene mobility (Fig 4D). In communities without prior exposure, plasmid-encoded mobile resistance genes increased background community stability relative to fully susceptible communities (estimated effect of resistance on stability, given mobility = 0.015 [0.010 to 0.020], Bayesian linear model BLM2; see S1 Text). And, consistent with this increased stability being driven by plasmid transfer into the background community, we observed significantly higher levels of mercury resistance in the background taxa following the perturbation when plasmids were present (Fig 4B and Fig F in S1 Text; effect of mobility on resistance frequency, given resistance = 0.21 [0.064 to 0.35], Bayesian linear model BLM3). In contrast, however, adding immobile chromosomal resistance genes strongly *reduced* background community stability (Fig 4D; effect of chromosomal resistance = −0.030 [−0.036 to −0.025, 95% CI], BLM 2). That is, in microbiomes harboring immobile resistance genes, background taxa were more strongly perturbed by the mercury pulse than in microbiomes entirely lacking resistance genes. Moreover, immobile resistance genes caused an even greater drop in background taxa stability when compared to communities harboring mobile resistance genes (effect of mobility on stability, given resistance = 0.046 [0.041 to 0.050, 95% CI], BLM2).

Together, these results support our predictions that resistance genes can have markedly different impacts on background community members, depending on their mobility.

Finally, we observed a striking difference in the response of the background and focal taxa to prior low-level mercury exposure. Specifically, when resistance genes were present within the community, prior exposure increased the stability of the background taxa (effect of exposure given resistance = 0.034 [0.019 to 0.048, 95% CI], BLM 2) yet decreased the stability of the focal taxon (interaction effect of focal community with exposure, given resistance = −0.17 [−0.19 to −0.14, 95% CI], BLM2). And, notably, this decreased stability was unique to resistant focal taxa, with prior exposure increasing the stability of the focal taxon if it was sensitive (interaction effect between focal community and exposure = 0.12 [0.10 to 0.14, 95% CI], BLM2). This suggests that, as predicted by our models, the increased stability of the background taxa might indeed be reducing the competitive release experienced by the resistant focal taxon during perturbations and thus decreasing the stability of the resistant focal taxon following prior exposure (cf. red region in Fig 3G). In contrast, the finding of prior exposure increasing sensitive focal taxon stability was likely due to the sensitive focal taxon already having been driven to low abundance by prior exposure, thus weakening the subsequent impact of the perturbation.

We did also observe some behavior not predicted by our model. In particular, in contrast to our modelling predictions, we found that prior exposure to mercury *increased* the stability of the background community in the presence of chromosomally encoded resistance genes (Fig 4D). This suggests mechanisms other than plasmid transfer, such as de novo mutation [31], may also be playing an important role in microbiome stability. Nonetheless, collectively, our experimental findings support our key theoretical predictions relating to the effects of resistance genes and their mobility upon the stability of competition-dominated microbiomes.

## Conclusions

Genes conferring resistance to stressors such as antibiotics, toxins, or pollutants are widespread within microbial communities, often encoded on mobile genetic elements such as plasmids or temperate phages [15–17]. While the consequences of resistance—particularly to antibiotics—for human health have been widely studied [32], the impact that resistance genes have on the structure and stability of microbial communities remains poorly understood. Here we combine novel theory and experiments to disentangle the diverse ways in which resistance genes influence the stability of microbiomes. Our work suggests that resistance genes typically increase overall community stability, particularly when encoded on highly mobile genetic elements. However, exactly how these genes influence microbiome properties depends upon the precise interplay between the properties of the gene and of the underlying microbiome. The same gene may have directly opposing effects upon microbiome stability, depending upon its mobility, or how individual taxa interact with one another. Moreover, not all taxa are affected equally, and, particularly in highly competitive microbiomes, the presence and transfer of resistance genes may benefit some taxa yet be detrimental to others.

The considerable variability in the effects of resistance genes within microbiomes introduces an interesting set of potential conflicts between genes, their bacterial hosts, and the broader ecosystem. For example, the spread of antimicrobial resistance poses a dangerous threat to public health [33]. Yet within a given microbiome, the spread of resistance into susceptible taxa may offer important ecological benefits, improving overall microbiome stability and protecting susceptible community members from outcompetition by resistant competitors. Increasing community stability through the spread of mobile resistance genes could also enable the maintenance of important ecosystem services within vulnerable microbial

communities. Similarly, transfer of a resistance gene into susceptible taxa may be advantageous for the fitness of the individual gene, increasing its frequency within the community. However, in certain microbiomes, spread of the resistance gene may be costly for the original host—reducing its advantage over otherwise susceptible competitors [34]. Which scenario plays out in any given system will depend upon the interplay between the ecological and evolutionary properties of the underlying microbiome and the nature of the environment these communities inhabit. Our study is far from exhaustive, and exploring how these eco-evolutionary dynamics play out between different scenarios such as rare versus common or static versus fluctuating perturbations—each of which have been shown to play important roles in shaping HGT in single species populations [35]—offers exciting future avenues for research. Taken together, however, the intricate dynamics revealed by our work underscore the important and sometimes complex effects mobile genetic elements can have upon microbial communities. Moreover, they also underline the importance of considering exactly how properties such as microbiome stability are quantified. Relying solely on coarse, whole-community metrics such as overall community abundances or dissimilarity may mask striking differences between individual community members and risks obscuring important eco-evolutionary dynamics.

To identify broad patterns in the impact of mobile resistance genes upon microbiomes, here we used relatively simple ecological models. The advantage of these simple models is that we can analyze large numbers of microbiomes in high-throughput. However, as a consequence, there are ecological and evolutionary features that we have not explicitly accounted for. For example, previous studies have suggested that plasmid transfer is more likely between closer phylogenetic relatives due to constraints on plasmid host range or spatial structuring [36]. Indeed, in our experiments, plasmid-mediated transfer of $Hg^R$ from *P. fluorescens* SBW25 appeared to be limited to congenerics, with more distantly related members of the community such as *B. megaterium* apparently unable to gain the resistance genes, suggesting they were unable to acquire or maintain either plasmid. Embedding simple constraints on conjugation within our model (Figs G-I in S1 Text) reveals that differences in transfer probabilities between taxa can subtly modulate the impact of mobile resistance genes upon community properties. Similarly, our model does not explicitly include de novo mutations, phenotypic plasticity, or social interactions [37] that may modulate how individual taxa respond to stressors without requiring the acquisition of resistance genes from other community members. Indeed, in our experiments, we observed an increase in the stability of background taxa after prior exposure to mercury within the chromosomally encoded resistance treatment (Fig 4D), suggesting mechanisms other than plasmid transfer may have driven this increased stability [31]. A large body of theoretical and experimental work has explored how HGT modulates the dynamics of single-species populations [35,38], and the challenge now is to understand how such species-level effects scale up to control the dynamics of complex microbiomes.

As our knowledge of the ubiquity of microbial communities has increased, so too has our desire to manipulate these microbiomes for our own benefit—using microbiome transplants or individual "probiotic" bacteria to supplant pathogens or simply to increase overall microbiome stability. Before we can engineer our microbiomes in a targeted manner, however, we need means of systematically understanding and predicting how these communities change over time. While some theoretical work has explored the impact of *ecological* processes in determining microbiome dynamics, until now, the impact of *evolutionary* processes such as HGT upon microbiome dynamics have been largely ignored. Our study shows that evolutionary changes can have profound and sometimes surprising impacts upon the ecological dynamics of microbial communities and that accounting for both ecological and evolutionary forces will be critical if we wish to fully understand and ultimately manipulate microbiome dynamics.

## Materials and methods

### Underlying microbiome model

In line with previous work [14,18], we model each microbiome as a network, where each node represents a taxon and each edge captures the interaction between them. However, now we extend this model to include 2 populations for each taxon, one with plasmids and one without [24]. For each taxon $i$, plasmid-free cells grow at a rate $r_i$ and plasmid-bearing cells grow at a rate $r_i − c$, where c indicates the cost of plasmid carriage. Plasmids can transfer within and between taxa, with the per cell rate of plasmid transfer from taxon $j$ into taxon $i$ defined as $\gamma_{ij} = \bar{\gamma} + \epsilon_{ij}$, where $\bar{\gamma}$ is the average plasmid transfer rate, and $\epsilon_{ij}$ is a noise term drawn from a Normal distribution to introduce variability in plasmid transfer rates between taxa (note, in instances where $\gamma_{ij} < 0$ we set $\gamma_{ij} = 0$). Finally, we introduce an inhibition term $−\beta D$, where D defines the level of antibiotic or toxin in the environment, and β the susceptibility of the population in question. More specifically, we set $\beta^s = 1$ for the susceptible population, and $\beta^r = 0.1$ for the resistant population, on the basis that even cells harboring the resistance gene will still be slightly affected by the stressor. Together, this enables us to define the growth rate of the plasmid-negative, $X^s$, and plasmid-positive, $X^r$, populations of a given taxon $i$ as

$$\frac{dX_i^s}{dt} = X_i^s \left( r_i - s_i(X_i^s + X_i^r) + \sum_{j=1,j\neq i}^{N} a_{ij}X_j^s + \sum_{j=1,j\neq i}^{N} a_{ij}X_j^r \right) - \sum_{j=1}^{N} \gamma_{ij} X_i^s X_j^r - \beta^s D X_i^s$$

$$\frac{dX_i^r}{dt} = X_i^r \left( (r_i - c) - s_i(X_i^s + X_i^r) + \sum_{j=1,j\neq i}^{N} a_{ij}X_j^s + \sum_{j=1,j\neq i}^{N} a_{ij}X_j^r \right) + \sum_{j=1}^{N} \gamma_{ij} X_i^s X_j^r - \beta^r D X_i^r$$

with equivalent expressions defining the dynamics of each other species $j = 2:N$ within the community (see Table 1 for parameter definitions and values). Notably, this model can also be extended to incorporate the phenomenon whereby plasmids can be lost during segregation at a rate δ; however, this does not qualitatively alter the results (Fig E in S1 Text).

Having established this new model, we generated a series of microbiomes, each composed of $N = 10$ taxa. Within any given microbiome, each taxon $i$ interacts with taxon $j$ with probability $C$. To investigate how interactions between taxa modulate the effect of resistance genes, we systematically alter the proportion of individual interaction terms, $a_{ij}$, that are positive, $P_m$, such that when $P_m = 0$ the community is purely competitive, when $P_m = 0.5$ the community

**Table 1. Parameter set used in main analysis.** Note random noise parameters (eg $\epsilon_{ij}$) are redrawn for each individual community, and changing these parameters does not qualitatively affect our results.

| Parameter | Value |
|---|---|
| Species number, N | 10 |
| Repeats | 100 |
| Interaction strength standard deviation, $\sigma$ | 0.015 |
| Self-inhibition, s | 0.1 |
| Probability of a given interaction, C | 0.7 |
| Resistance cost, c | 0.005 |
| Average plasmid mobility, $\bar{\gamma}$ | $0.5*10^{-3}$ |
| Per species pair conjugation noise term, $\epsilon_{ij}$ | $\bar{\gamma} * 0.1*N(0,1)$ |
| Mercury, D, perturbation | 0.1 |
| Mercury, D, prior exposure | 0.01 |
| $\beta$, susceptible | 1 |
| $\beta$, resistant | 0.1 |

contains a mixture of all interaction types, from competitive and ammensal, through exploitation, to cooperative and commensal, and when $P_m = 1$ the community is purely cooperative. Finally, we choose the magnitude of each $a_{ij}$ from a half-normal distribution with standard deviation $\sigma = 0.015$, and, in line with previous work, we set the intrinsic growth rates $r_i$ such that in the absence of any stressor, the community has a linearly asymptotically stable equilibrium at $X_i^* \approx 1 \forall i \in 1, \dots, N$. Importantly, setting the intrinsic growth in this manner introduces a trade-off in growth strategies, with species investing either in their own intrinsic growth rate or growth via cooperation with others, reducing unrealistic explosive behavior in highly cooperative communities.

To allow comparison to previous work, throughout, we choose a standard set of ecological parameters for our model, then select our evolutionary parameters to scale accordingly. For example, we vary our rate of plasmid transfer gamma across $[10^{-6}, 10^{-2}]$, covering scenarios from effectively immobile genes, through those with an equivalent probability of intertaxa plasmid acquisition to that observed in [24], to rare "super-mobile" plasmids. Importantly, however, we find qualitatively equivalent results when varying each of these key parameters (Figs C-E and J in S1 Text).

## Quantifying microbiome stability

We calculate the stability of any given community by simulating its dynamics in response to a perturbation. Specifically, we first solve the community dynamics for an initial "adjustment period" of t = 500 time units, allowing the community to reach an approximate steady state in terms of taxa abundances (importantly, while typically also effectively stable, resistance gene frequencies may not be at a true equilibrium). We then briefly perturb the community by setting the stressor level within the environment to D = 0.1 for t = 25 time units then measure the difference in abundance of each taxon $i$ before, $A_i^b$, and after, $A_i^a$, this perturbation. We define the stability of each individual taxon $i$ based on their fold change in response to the perturbation, $Stability_i = \log_{10} \frac{A_i^a}{A_i^b}$, such that more negative values indicate less stable taxa (for simplicity, we set values of Stability greater than zero to zero, although dropping this step does not qualitative change our results). We then define the stability of the whole community (or background community), as the mean of $Stability_i$ across all taxa (or across only the background taxa)—a measure often referred to as the Robustness of a community.

To explore the impact of resistance genes on stability for each community, we perform these simulations when all taxa are susceptible to the stressor and when one randomly chosen taxon harbors a mobile resistance gene. To explore the impact of low-level selection on stability, we perform these simulations when the community initially adjusts in a stressor-free environment and when the community adjusts in the presence of a low-level of the stressor (setting D = 0.01 during the adjustment period), with all simulations performed in MATLAB R2020a.

## Strains and culture conditions

To test the accuracy of our predictions, we assembled experimental model microbiomes, composed of a defined background community augmented with a predetermined focal taxon. For our focal species, we used *Pseudomonas fluorescens* SBW25 labelled with a gentamicin resistance marker using the mini-Tn7 transposon system [24,31,39,40]. More specifically, we generated independent *P. fluorescens* strains that were either susceptible to mercury, or harbored a mercury resistance gene, Hg$^R$, on either the chromosome, the conjugative plasmid pQBR57 [40,41], or the conjugative plasmid pQBR103 [41]. Individual colonies of each taxa (one for

each replicate) were isolated on selective KB agar and grown overnight at 28 degrees in 6 ml KB broth (10 g glycerol, 20 g proteose peptone no. 3, 1.5 g $K_2HPO_4 \cdot 3H_2O$, 1.5 g $MgSO_4 \cdot 7H_2O$, per litre; [42]) in a shaking incubator.

To generate our background community, we plated supernatant from non-autoclaved John Innes No.2 potting soil on nutrient agar plates, which were then grown for 48 hours in a 28-degree centigrade incubator. Following this incubation period, we randomly selected 96 colonies, which we screened against $Hg^{2+}$ and Gm to ensure no community members already harbored phenotypic resistance to either our stressor or selective marker. Each of these bacterial taxa were grown separately overnight in 6 ml KB broth in a shaking incubator and then washed and mixed at an equal volume to make the background community. To generate each final microbiome community, we resuspended an equal volume of the background community and an overnight culture of *P. fluorescens* in M9 buffer. We then diluted this suspension by 1:10 with M9 buffer and used 100 μl to initiate each experimental replicate. Populations were cultured in sterile soil microcosms consisting of 10 g twice autoclaved John Innes No.2 potting soil supplemented with 900 μl of sterile $H_2O$ and maintained at 28 degrees at 80% relative humidity. This medium retains a soil-like texture and spatial structuring and is not a slurry [24,41].

## Community perturbation experiment

We established 12 replicate communities per SBW25 genotype: SBW25, SBW25 with chromosomal $Hg^R$, SBW25 carrying $Hg^R$ encoded on pQBR57, SBW25 carrying $Hg^R$ encoded on pQBR103. These were propagated by serial transfer of 1% of the community every 4 days into fresh sterile potting soil microcosms for 12 transfers following a previously established serial transfer protocol [24]. Specifically, for each community at each serial transfer, 10 ml of M9 salts solution and 20 sterile glass beads were added to each microcosm prior to vortexing for 1 minute to resuspend bacterial cells into the supernatant of which 100 μl was then transferred to initiate a fresh microcosm. In half of these lines, communities were supplemented with mercuric chloride (at 16 μg/g $HgCl_2$) from transfer 2 onwards, allowing us to capture communities with and without prior stressor exposure. At transfer, 10 all communities were perturbed by exposure to 128 μg/g $HgCl_2$ and then propagated for a further serial transfer at 0 μg/g $HgCl_2$. Samples of each community were cryogenically stored at each serial transfer in 20% glycerol. Throughout the experiment, we determined the population density of SBW25 by diluting and plating samples onto KB agar supplemented with 6 μg/ml gentamicin, and the abundance of the entire community by plating onto nutrient agar. In addition, we determined the frequency of the $Hg^R$ phenotype in the community as a whole by plating onto nutrient agar supplemented with 20 μM $HgCl_2$, while we determined the frequency of $Hg^R$ resistance in the focal strain by plating onto nutrient agar supplemented with 20 μM $HgCl_2$ and 6 μg/ml gentamicin.

## 16S rRNA gene sequencing and analysis

To quantify community composition, we extracted whole community DNA samples from the thawed stocks stored on days 10 and 11 (that is, before and after the mercury shock; Fig 4A). Specifically, we extracted DNA using QIAGEN DNeasy PowerSoil kits, following manufacturer instructions with the exception that stocks were initially spun down and resuspended in 1× M9 to remove glycerol. DNA concentrations were assessed using Qubit fluorometer 3.0 (Thermo Fisher Scientific) and diluted to 20 ng $μl^{-1}$ where possible before samples were sent for downstream PCR amplification of the v4 region of 16s rRNA gene and sequencing on the Illumina MiSeq platform. The raw forward and reverse reads were merged and processed using QIIME1 [43]. Reads were stripped of their primer and barcoding sequences using

Cutadapt [44] and untrimmed reads were discarded. Reads were truncated at 254 bp (size of the amplicon). Reads were then dereplicated using Vsearch [45] and clustered into OTUs using Usearch [46] with 97% similarity. OTUs were then filtered based on OTUs that appeared in the positive control (14 OTUs in total). Putative taxonomic identification of the 14 OTUs was performed using BLAST [47] to align the OTU sequence data to the NCBI nucleotide database, listed in Table A in S1 Text. Total abundances of the focal species and the background community were determined by multiplying the relative abundances of each with the total community abundances calculated by CFU counts.

## Statistics

Differences in microbiome stability and resistance frequency were estimated using 3 Bayesian linear models, accounting for the experimental structure (where relevant, the presence of resistance, its mobility, foreground versus background communities, prior exposure to low-level mercury selection, and timing of measurement relative to mercury shock), nonhomogeneity of variances and, where appropriate, nonnormality of residuals, using broad priors. These models were fitted with the brms package [48] (version 2.16.3), which uses STAN via the R language [49] (version 4.1.2). Four MCMC chains were used, each of 4,000 iterations, where the first 2,000 were discarded as warm-up, resulting in 8,000 draws from the posterior distribution. Convergence was checked visually using plots of the draws and via the R-hat value [50], which will equal 1 at convergence and was 1.0 for all parameter estimates reported in the main text and supplemental analyses. All values are reported as a mean with 95% credibility interval (CI). Details of model structures and estimated parameters for each of the 3 models (Tables B-D in S1 Text) are given in the Supporting information.

## Supporting information

**S1 Text. Supporting text and figures, including Supporting Tables A-D and Figs A-J.** (DOCX)

## Acknowledgments

We thank D.R. Gifford and W.P.J. Smith for helpful discussions.

## Author Contributions

**Conceptualization:** Katharine Z. Coyte, Michael A. Brockhurst.

**Data curation:** Katharine Z. Coyte, Michael A. Brockhurst.

**Formal analysis:** Katharine Z. Coyte, Christopher G. Knight, Michael A. Brockhurst.

**Funding acquisition:** Katharine Z. Coyte, Cagla Stevenson, Ellie Harrison, James P. J. Hall, Michael A. Brockhurst.

**Investigation:** Katharine Z. Coyte, Cagla Stevenson, Christopher G. Knight, Ellie Harrison, James P. J. Hall, Michael A. Brockhurst.

**Methodology:** Katharine Z. Coyte, Cagla Stevenson, Christopher G. Knight, Ellie Harrison, James P. J. Hall, Michael A. Brockhurst.

**Project administration:** Katharine Z. Coyte, Michael A. Brockhurst.

**Resources:** Katharine Z. Coyte, Michael A. Brockhurst.

**Software:** Katharine Z. Coyte, Michael A. Brockhurst.

**Supervision:** Katharine Z. Coyte, Michael A. Brockhurst.

**Validation:** Katharine Z. Coyte, Michael A. Brockhurst.

**Visualization:** Katharine Z. Coyte, Michael A. Brockhurst.

**Writing – original draft:** Katharine Z. Coyte, Michael A. Brockhurst.

**Writing – review & editing:** Katharine Z. Coyte, Cagla Stevenson, Christopher G. Knight, Ellie Harrison, James P. J. Hall, Michael A. Brockhurst.

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
