## [Editor Report · Decision Letter 0]

21 Jun 2022

Dear Dr. Coyte, 

Thank you for submitting your manuscript entitled "Horizontal gene transfer and ecological interactions jointly control microbiome stability" for consideration as a Research Article by PLOS Biology.

Your manuscript has now been evaluated by the PLOS Biology editorial staff and I am writing to let you know that we would like to send your submission out for external peer review.

Once your full submission is complete, your paper will undergo a series of checks in preparation for peer review. After your manuscript has passed the checks it will be sent out for review. To provide the metadata for your submission, please Login to Editorial Manager (https://www.editorialmanager.com/pbiology) within two working days, i.e. by Jun 23 2022 11:59PM.

Kind regards,

Paula

---

Senior Editor

PLOS Biology

---

## [Decision Letter · Decision Letter 1]

10 Aug 2022

Dear Dr. Coyte,

Thank you for your patience while your manuscript "Horizontal gene transfer and ecological interactions jointly control microbiome stability" went through peer-review at PLOS Biology. Your manuscript has now been evaluated by the PLOS Biology editors, an Academic Editor with relevant expertise, and by several independent reviewers. One of the reviewers evaluated the previous version of your manuscript. 

In light of the reviews, which you will find at the end of this email, we are pleased to offer you the opportunity to address the comments from the reviewers in a revision that we anticipate should not take you very long. In particular, reviewer #1 requests a few clarifications, and reviewer #2 for additional discussion of the health implications of the current work. We consider that you should respond to these comments, and, if possible, identify the most apparent (potential) health implications in the form of a short additional paragraph.

We will assess your revised manuscript and your response to the reviewers' comments with our Academic Editor aiming to avoid further rounds of peer-review, although might need to consult with the reviewers, depending on the nature of the revisions.

**IMPORTANT - SUBMITTING YOUR REVISION**

*Resubmission Checklist*

*Published Peer Review*

*PLOS Data Policy*

*Blot and Gel Data Policy*

Sincerely,

Paula

---

Senior Editor

PLOS Biology

REVIEWS:

Reviewer #1: Plasmids in bacteria and Plasmid-Host Interaction.

Reviewer #2: Epidemiology and evolutionary dynamics of antibiotic treatment and resistance.

Reviewer #1: The authors thoroughly responded to each comment and adjusted the manuscript accordingly. I apologize for some of my earlier misunderstandings, but I must say that the manuscript is more clear now with the additional clarifications. 

I have just a few last minor comments:

1. L. 413: "...16S rDNA gene amplicon sequencing": This should really read 'by amplicon sequencing of the 16S rRNA gene'. (There is no ribosomal DNA).

2. L. 418: "After adjusting to microcosm conditions, each microbiome contained between three and five taxa at appreciable abundances". Does this mean that more than 90 of the added strains were no longer detectable? That is worth pointing out I think. 

It would also be good to quantify 'appreciable abundances": what is the expected detection limit of the amplicon sequencing?

3. L. 451. Descriptions of Fig. 4C, D: I still think that there are a few experimental observations that are not entirely expected. The paper would benefit from pointing those out right away, in addition to pointing out the observations that are in line with the expectations. I think I see two or three:

a) "We also observed a positive impact of gene mobility on overall microbiome stability, however, this effect was small (estimated effect of mobility on stability, given resistance = 0.0065 [0.0017 - 0.011, 95% CI], BLM1), AND DID NOT CHANGE SIGNIFICANTLY WITH PRIOR LOW-LEVEL EXPOSURE TO MERCURY": If I understood the model output well, the latter points were not entirely expected based on the model: gene mobility had a large effect on total stability and prior exposure was supposed to select for horizontally transferred resistance genes and thus have a larger positive effect on stability. Is it because overall stability is driven by that of the focal species and 'hides' the subtle changes in stability of the background?

b) Fig 4D, E: the fact that the background community benefited from prior exposure (by quite a lot: orange dot vs blue dot) when the resistance gene was chromosomal (Fig 4D), and that the sensitive focal strain benefited from prior exposure (by quite a lot) needs some explanation. The authors bring some of this up in the Conclusions but it is puzzling the reader right here (around L. 497) when looking at Fig 4.

4. L. 778: "These were propagated by serial transfer of 1% of the community every 4 days into

fresh soil microcosms for twelve transfers. It would help to understand that (i) this was sterilized soil, (ii) was it actually potting soil, or a slurry? (iii) When 'transferring 1%': is it 1% of the mass of the soil, or volume of soil or slurry? 

Reviewer #2: This manuscript is important to academic microbial ecology and, at the same time, directly relevant to human health. As the authors note and the editors agree, we know relatively little about the mechanisms determining the structure of microbial communities (microbiomes), their strain and species composition, the relative densities of bacteria in these communities, and the factors limiting their total densities and maintaining the stability of these communities.   This study focuses primarily on the mechanisms responsible for the stable maintenance of these communities when they are confronted with stresses that kill and prevent the replication of the bacterias. From a human health perspective, the response to stress, like antibiotics, is particularly critical. Disruption of the gut microbiome stability leads to antibiotic-associated pseudomembranous colitis, which can have serious health and even lethal consequences.  

The perspective presented in this report, that genes that code for resistance to the stresses and especially those that are horizontally transmitted by conjugative plasmids and temperate bacteriophage, is, to my knowledge, novel. Their model seems to provide a theoretical basis for this proposition, and their soil microbiome experiments support it.   I also appreciate the combination of theory and experiments.  

I support and recommend publishing this report in PLoS Biology, but only if the revised version of this report meet the following criteria:

I agree with almost all the extensive comments of the three editors that reviewed this report. For this paper to be acceptable for publication, these editors should tell PLoS Biology that they agree with the author's response to their reviews and the modification made in the manuscript to address their concerns. 

pt.All three editors raised concerns about the criteria the authors use for the stability of the microbiome. I have concerns about that as well. I will be satisfied if the "editors" agree with how this issue is addressed in the current incarnation of this 

report.

In revising this manuscript, the authors should.

Consider the implications of this study to our understanding of the mechanisms responsible for maintaining the stability of the human gut microbiome.

Consider why stress-mediated disturbances of the human gut microbiome can manifest as a disease. 

Briefly discusses that the gut microbiome disturbances resulting in pseudomembranous colitis can be successfully treated with fecal microbiome transplants.  

Discuss the implications of horizontal transfer of antibiotic resistance genes, which this study considers to be responsible for maintaining the stability of the genome to the generation and dissemination of antibiotic-resistant pathogens.

Comment on the hypothesis that bacteriophages play a prominent role in determining the distribution and abundance of different species and strains of bacteria in the human microbiome.

---

## [Editor Report · Decision Letter 2]

12 Sep 2022

Dear Dr Coyte,

Thank you for your patience while we considered your revised manuscript "Horizontal gene transfer and ecological interactions jointly control microbiome stability" for publication as a Research Article at PLOS Biology. This revised version of your manuscript has been evaluated by the PLOS Biology editors and the Academic Editor.

Based on our Academic Editor's assessment of your revision, we are likely to accept this manuscript for publication, provided you satisfactorily address the following data and other policy-related requests.

DATA POLICY:

Regardless of the method selected, please ensure that you provide the individual numerical values that underlie the summary data displayed in the following figure panels as they are essential for readers to assess your analysis and to reproduce it: Figures 2BCD, 3BCDEFG, 4BCDE and Supplementary Figures S1ABCD, S2ABCDEF, S3ABCDEFGHIJKL (Please note that in the figure the panels are named A-F twice and the figure legend indicates A-F and G-L), S4ABCDEFGHIJKL (Same problem as in S3), S5ABCDEF, S6, S7, S8, S9, S10.

**Please also ensure that figure legends in your manuscript include information on where the underlying data can be found, and ensure your supplemental data file/s has a legend.**

We expect to receive your revised manuscript within two weeks.

*Published Peer Review History*

*Press*

Sincerely,

Paula

---

Senior Editor,

pjaureguionieva@plos.org,

PLOS Biology

---

## [Editor Report · Decision Letter 3]

23 Sep 2022

Dear Dr. Coyte,

Thank you for the submission of your revised Research Article "Horizontal gene transfer and ecological interactions jointly control microbiome stability" for publication in PLOS Biology. On behalf of my colleagues and the Academic Editor, Arjan de Visser, I am pleased to say that we can in principle accept your manuscript for publication, provided you address any remaining formatting and reporting issues. These will be detailed in an email you should receive within 2-3 business days from our colleagues in the journal operations team; no action is required from you until then. Please note that we will not be able to formally accept your manuscript and schedule it for publication until you have completed any requested changes.

PRESS

Sincerely, 

Paula 

---

Senior Editor

PLOS Biology
